# Albumin-Based Zn (II)-Quercetin Enzyme Mimic Scavenging ROS for Protection against Cardiotoxicity Induced by Doxorubicin

**DOI:** 10.3390/ph15121524

**Published:** 2022-12-08

**Authors:** Zehua Shao, Ran Li, Dongxing Shao, Hao Tang, Yu Han

**Affiliations:** 1Heart Center of Zhengzhou University People’s Hospital, Fuwai Central China Cardiovascular Hospital of Zhengzhou University, Henan Provincial People’s Hospital, Zhengzhou 450003, China; 2National Health Commission Key Laboratory of Cardiovascular Regenerative Medicine, Fuwai Central China Cardiovascular Hospital & Central China Branch of National Center for Cardiovascular Diseases, Zhengzhou 451464, China; 3Beijing Key Laboratory of Protein Posttranslational Modifications and Cell Function, Department of Biochemistry and Molecular Biology, School of Basic Medical Sciences, Peking University Health Science Center, 38 Xueyuan Road, Beijing 100191, China

**Keywords:** albumin-based enzyme mimic, quercetin, doxorubicin, cardiotoxicity, ROS

## Abstract

Doxorubicin (DOX) is a chemotherapeutic agent that can cause cardiotoxicity leading to progressive, chronic, life-threatening cardiomyopathy, called DOX-induced cardiomyopathy (DIC). DIC is a fatal cardiomyopathy with a worse prognosis compared to other cardiomyopathies and limits the use of DOX in malignancies due to its cardiotoxicity. DIC has been proven to be associated with reactive oxygen species (ROS)-induced side effect damage in cardiac myocytes. Currently, scavenging of reactive oxygen species is a practical strategy to reduce chemotherapy-associated DIC. Although quercetin has already been reported to have superior antioxidant activity, its clinical application is severely limited due to its rapid degradation and poor tissue absorption. Herein, we reported the preparation of a novel enzyme mimic via coordinated albumin, Zinc Ion (Zn^2+^) and quercetin. The enzyme mimics were capable of simultaneously increasing the biocompatibility and efficiently overcame the drawbacks of free quercetin, and were achieved by long circulation in vivo. Most importantly, these quercetin-based enzyme mimics had no effect on the antioxidant activity of quercetin. These beneficial therapeutic properties, together with high drug-carrying capacity and redox stimuli, will significantly improve quercetin’s alleviation of chemotherapeutic cardiotoxicity without causing significant side effects. Therefore, nanoparticles of albumin-based Zn (II)-Quercetin have a promising clinical application as an effective agent for mitigating the cardiotoxicity of chemotherapy.

## 1. Introduction

Although cancer-related mortality has decreased in the last 20 years, cancer remains the leading cause of death worldwide [1]. Drug development technology has significantly improved cancer patient survival rates; however, many chemotherapy drugs are capable of causing adverse drug reactions, with cardiovascular toxicity being one of the most common and life-threatening adverse reactions [2,3]. As cancer survival rates improved, the prognosis of patients with cancer, and anti-cancer drugs that lead to an increased risk of cardiovascular problems, quickly attracted attention, including tumor treatment-related cardiac damage as the most common side effect of chemotherapy. In about 10% of patients with tumors, anthracycline-based preparations are commonly used in many malignant diseases and have a distinct curative effect from first-line chemotherapy drugs, such as doxorubicin and amrubicin [4,5,6]. Since the introduction of doxorubicin (DOX), the most widely used and effective anthracycline-based solid tumor chemotherapy for hematopoietic system tumors, cancer patient survival rates have increased dramatically [7].

DOX is a type of cell cycle nonspecific anthracycline-based chemotherapy drug that works by inhibiting DNA transcription and replication to increase oxidative stress in tumor cells [8,9]. In clinical practice, it is commonly used as the first-line drug of choice, either in combination with other antitumor drugs or in combination with surgery and radiotherapy to eradicate tumors. However, the killing effect of doxorubicin is non-selective, and due to its myocardial affinity, doxorubicin has particularly pronounced toxic side effects on the heart, frequently demonstrating concentration-dependent cumulative and potentially fatal myocardial injury ultimately leading to severe cardiomyopathy and heart failure [10]. The biological mechanisms by which DOX causes myocardial toxicity have not been fully understood despite decades of research. As a result, the core DOX toxicity mechanism may be intricate and multifaceted [11]. Currently, it is generally accepted that the primary causes of DOX heart toxicity are oxidative stress, a disorder of Ca^2+^ metabolism, and metabolic damage. The main features of ferroptosis (iron death) are intracellular iron accumulation and decompensated lipid peroxidation [12,13]. The typical microstructure characteristics include glutathione (GSH) depletion and increased intracellular reactive oxygen species, as well as mitochondrial swelling, increased membrane density, and decreased or absent mitochondrial kurtosis (ROS) [14,15,16]. Although there is still much debate and discussion surrounding the pathogenic mechanism and pathophysiological changes, it is generally accepted that the excessive production of ROS is the primary cause of the myocardial injury brought on by DOX. Decreased doxorubicin-induced toxicity is therefore crucial for enhancing cancer patients’ survival and quality of life [12,17,18].

Quercetin, a dietary flavonoid derived from organic plants, is frequently used in the treatment of a variety of inflammatory diseases [19,20,21] due to its superior antioxidant and anti-inflammatory activity. However, due to insufficient stability or low tissue exposure levels, these have failed in clinical trials. The development of a novel quercetin nanomedicine that increases cardiac accumulation while also enhancing hydrolytic stability is crucial to facilitating the clinical application of quercetin. According to reports, various delivery techniques have been used to reduce drug toxicity and enhance drug stability [22,23,24]. Among them, protein-based drugs have gained huge attention due to their high biosafety and cellular uptake ability [25,26]. In the current study, we used a bovine serum albumin (BSA) driven co-assembly strategy to create supramolecular quercetin nanodrugs. On the basis of coordination and numerous noncovalent interactions, well-defined and homogeneous quercetin nanomedicines (Que NPs) were created by using metal-bound amino acids and quercetin as building blocks. Through metal coordination and molecular stacking, Que NPs showed significant enhanced stability in the neutral, physiological media. More importantly, this assembly process had no impact on quercetin’s capacity to scavenge free radicals. Encouraged by these excellent properties, we systematically evaluated the efficacy of Que NPs scavenging free radicals against cardiotoxicity induced by doxorubicin through cellular and animal models.

## 2. Results and Discussion

### 2.1. Synthesis and Characterization of Que NPs

The detoxification repair agent Que NPs was fabricated via BSA coordinated co-assembly with Quercetin and Zn^2+^. Transmission electron microscopy (TEM) images of the samples show the formation of spherical nanostructures (Figure 1), which are 80~130 nm in diameter. Dynamic light scattering (DLS) revealed that Que NPs has an average diameter of 84 nm and consistent with TEM results. Furthermore, the superoxide dismutase enzyme-like (SOD) activities of free Que and Que NPs were studied under physiological conditions using the xanthine oxidase/cytochrome C system as a model. As shown in Figure 1C, free Que and Que NPs all showed SOD activities in phosphate buffer (pH 7.4), and in both, the final inhibition ratio was close to 100%. These findings demonstrated that Que NPs possess SOD-like activity, which may be advantageous for enhancing the therapeutic efficacy for inflammation-related diseases. As shown in Figure 1D, the stability of quercetin can be significantly improved by forming quercetin-Zn^2+^ complexes. Importantly, the stability of quercetin was further improved by the formation of Que NP, even after being placed under physiological conditions for 72 h, there was still a druge residue rate of approximately 60%. These results indicate that Que NPs showed excellent dispersion, homogeneity, stability and potential tissue penetration. Therefore, we have reason to believe that such Que NPs can be better used in biomedical applications.

### 2.2. Protection against DOX-Induced Oxidative Stress In Vitro

We then examined whether the Que NPs possessed a good ROS scavenging property for protecting cells from oxidative damage, which was prompted by their excellent free radical scavenging ability. First, we verified the biocompatibility of Que and Que NPs. The result showed that Que NPs exhibited a certain promotional effect on cell proliferation (Figure 2A). Under the simulation of DOX injury, a cell-counting life test and revealed that Que NPs had a visible protective effect on AC16 myocardial cells (Figure 2B). The ROS generation of AC16 cells was revealed by the DCFH-DA ROS probe. As shown in Figure 2C, after being DOX-induced, the ROS level of AC16 cells was significantly enhanced. However, the cells treated with Que NPs displayed a significant drop in ROS levels.

The Bcl-2 gene (B cell lymphoma/leukemia-2 gene) is an oncogene that promotes cell division and proliferation while inhibiting cell apoptosis, and caspase-3 is an apoptotic protein that plays a critical role in the initiation and regulation of cell apoptosis. The detection of differences in its expression can also demonstrate the protective effect of Que NPs on cell apoptosis.

Therefore, the in vitro-cultured AC16 cells were divided into four groups: one group received the vehicle, one group received DOX, one group received Que NPs, and the final group received both DOX and Que NPs. Western blotting was used to detect the expression levels of caspase-3 and Bcl-2 following cell lysis to determine the relationship between DOX and apoptosis induction. As shown in Figure 3A–C, cells treated with DOX exhibited a significant increase in apoptosis. At the protein level, the expression of caspase-3 was significantly increased, whereas the expression of pro-caspase-3 was decreased, indicating that DOX activated the apoptosis pathway and promoted the conversion of more caspase-3 precursors into caspase-3. Apoptosis was induced and the expression of the oncogene Bcl-2 was significantly decreased, the protective mechanisms of apoptosis were eliminated, and apoptosis was significantly triggered. In general, DOX activated caspase-3 and Bcl-2 pathways to induce cardiomyocyte injury and death. When DOX and Que NPs were administered concurrently, the expression of caspase-3 was significantly inhibited, which was not significantly different from the expression in the control group, and the expression of Bcl-2 was high, indicating that the cell growth status was normal, and the injury of cardiomyocytes was treated and improved. It is worth noting that both the vehicle and Que NPs exhibited negligible cytotoxicity in this in vitro experiment, almost causing cell death, and that the content of Bcl-2 increased slightly after quantitative detection after the addition of Que NPs, indicating that the Que NPs had the ability to promote cell growth. This demonstrated that the nanocomplex we created is safe and efficient, with promising applications in the fight against myocardial injury. PI probe and Annexin V-FITC staining were used in flow cytometry to detect the degree of apoptosis. According to the results shown in Figure 3D and Appendix A, the apoptosis rate of myocardial cells in DOX model mice was 24.1%, which was extremely high. After treatment with Que NPs, the apoptotic rate of cardiomyocytes was significantly reduced to 16.89%, with a decrease rate of approximately 30.04%, indicating that Que NPs had a good protective effect.

### 2.3. Protection against DOX-Induced Oxidative Stress In Vivo

The in vivo efficacy of Que NPs was then determined using DOX model mice [27,28]. Figure 4 depicts the overall design of the animal study. Que NPs were injected every other day from the start of the experiment until the fourth day when the doses were reduced to once a week until the mice had received a total of six doses before being sacrificed. On the sixth day and then once a week, DOX at a dose of 5 mg/kg was injected into the DOX model mice. After four injections in total, echocardiography was performed at an interval of one week to observe the myocardial changes and cardiac function of the mouse model. The heart and blood samples were then gathered for immunohistochemistry staining and TdT-mediated dUTP nick-end labeling (TUNEL). EF%, FS%, Lvid.s (left ventricular systolic diameter), and Lvid.d (left ventricular diastolic diameter) are reliable indicators of left ventricular systolic function, and there is evidence that their values decreased significantly in the DOX treatment group. As depicted in Figure 4A,B, we can see significantly decreased EF% and FS% values indicated severe cardiac dysfunction, in the Figure 4A,B we can see that DOX administration groups EF% and FS% values have been reduced to 25–39% and 12–19%, respectively. As show in Figure 4C–F, Lvid.s (left ventricular systolic diameter) and Lvid.d (left ventricular diastolic diameter) in mice receiving regular injections of DOX were significantly increased by even more than twofold compared to the control group, according to M type echocardiography results, representing diastolic and systolic heart enlargement, decreased cardiac muscle contraction ability and compliance. The EF, FS, Lvid.s, and Lvid.d were significantly decreased in the Que NP-treated group. In conclusion, DOX-treated mice are susceptible to impairments in myocardial systolic function, pumping ability, long-term ventricular remodeling and hypertrophy, and eventual heart failure. Notably, the Que NPs-DOX treatment significantly reduced the levels of some indicators. There was even complete prevention of myocardial damage by DOX, with no significant difference compared to the control group, which is extremely encouraging and suggests that Que NPs have irreplaceable effects on the protection of cardiac function.

The detection results of myocardial injury markers such as creatine kinase MB (CK-MB), serum cardial troponin T (cTnT), lactate dehydrogenase (LDH), and B-type natriuretic peptide (NT-proBNP) are depicted in Figure 5A–D, indicating that the number of myocardial injury markers increased significantly in DOX model mice, which shows that the mouse myocardium was severely damaged during the one-month administration. However, the addition of Que NPs significantly improved the condition of myocardial injury, and the values of various indexes decreased significantly, which was comparable to the control group, indicating that Que NPs played a significant role in reducing DOX-induced myocardial injury, preventing ventricular evil remodeling, and protecting cardiac function.

TUNEL staining was used to determine the apoptosis ratio of murine heart tissues (Figure 6A and Appendix A), and the results showed that apoptosis was significantly increased in DOX treatment groups. Similarly, to the in vitro results, the Que NP treatment group significantly reduced the apoptosis ratio, implying that Que NPs could significantly increase the survival ratio of heart cells in the infarct area after DOX treatment. All these results provide evidence that the Que NPs can significantly act against cardiotoxicity induced by doxorubicin through scavenging ROS. To evaluate the potential biomedical applications of Que NPs in vivo, we assessed the biosafety of the Que NP-treated major organs (heart, liver, spleen, lung, kidney) in mice. H&E staining results suggested that sections of major organs after Que NPs administration exhibited no inflammation or other pathological changes (Figure 6B).

## 3. Materials and Methods

Detailed materials and methods are described in the ‘Appendix A’. The AC16 human cardiomyocyte cell, a proliferating cell line that was derived from the fusion of primary cells from adult human ventricular heart tissues with SV40 transformed, was purchased from the company Merckmillipore (https://www.merckmillipore.com/CN/zh/product/AC16-Human-Cardiomyocyte-Cell-Line,MM_NF-SCC109; accessed on 2 December 2021). All animal experiments were approved by the Institution Animal Ethics Committee of Zhengzhou University (license No. 2021042501). Experimental procedures were performed in compliance with the Guidelines for the Care and Use of Laboratory Animals published by the United States National Institutes of Health (NIH Publication, revised 2011). Mice were given adaptive feeding for one week and were conducted in subsequent experiments.

## 4. Conclusions

To summarize, DOX is widely used as a first-line antitumor chemotherapy drug and has a significant therapeutic effect in a variety of cancer types. DOX, on the other hand, causes abnormal mitochondrial membrane potential in cardiomyocytes and generates a large number of reactive oxygen species, resulting in long-term and slow damage to cardiomyocytes. Que NP complexes have the ability to resist reactive oxygen species and have a preventive effect, thereby protecting cardiomyocytes, delaying ventricular remodeling, and maintaining cardiac function stability. The experiments showed that the structure and properties of Que NP complexes are stable and that they can function in a variety of pH, temperature, and ion environments for an extended period of time. The biomimetic mineralized Que NP complex carrier has a high loading efficiency, low cytotoxicity, high efficiency, and rapid cell uptake, making it an excellent candidate for application. It offers a novel approach to the prevention and treatment of doxorubicin-induced myocardial injury and brings good news to more cancer patients who cannot tolerate the side effects of chemotherapy.

## Figures and Tables

**Figure 1 pharmaceuticals-15-01524-f001:**
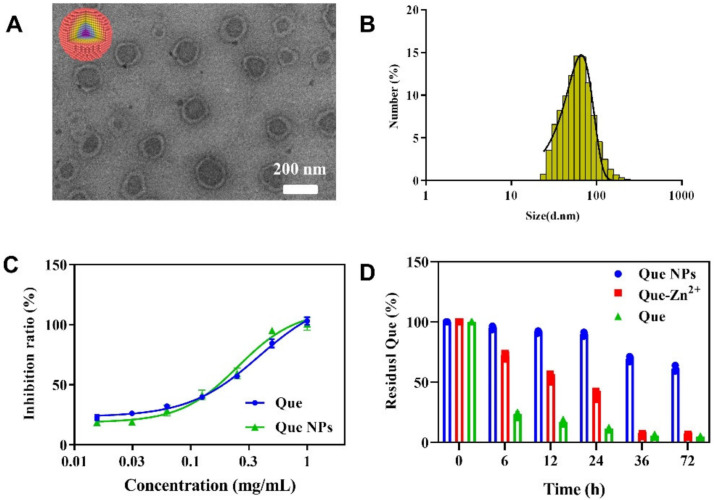
Characterization of Que NPs nanomedicine: (**A**) TEM image of Que NPs nanoagents; (**B**) diameter of Que NPs; (**C**) the SOD-like activities of free Que and Que NPs; and (**D**) degradation profiles of quercetin in various formulations under physiological conditions (pH 7.4, *n* = 3).

**Figure 2 pharmaceuticals-15-01524-f002:**
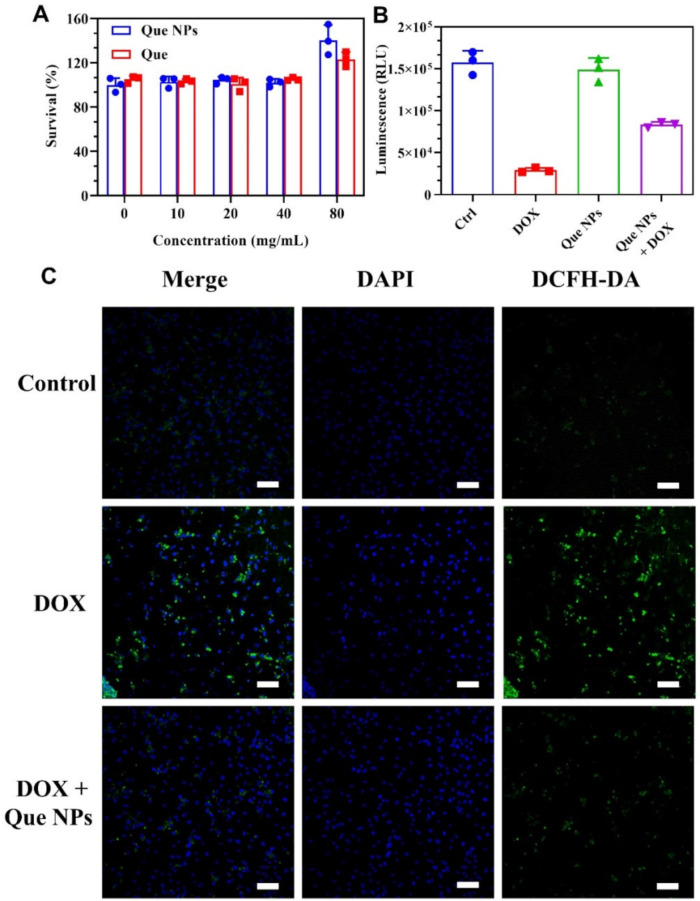
Assessment of the in vitro biocompatibility and protective effect of Que NPs on AC16 cells: (**A**) the cytotoxicity of AC16 cells with different concentrations of Que NPs was assessed by MTT after 24 h of culture; (**B**) the protective effect of AC16 cells with Que NPs was assessed by cell-counting life; and (**C**) the level of ROS was measured by DCFH-DA probe (scale bar, 100 μm).

**Figure 3 pharmaceuticals-15-01524-f003:**
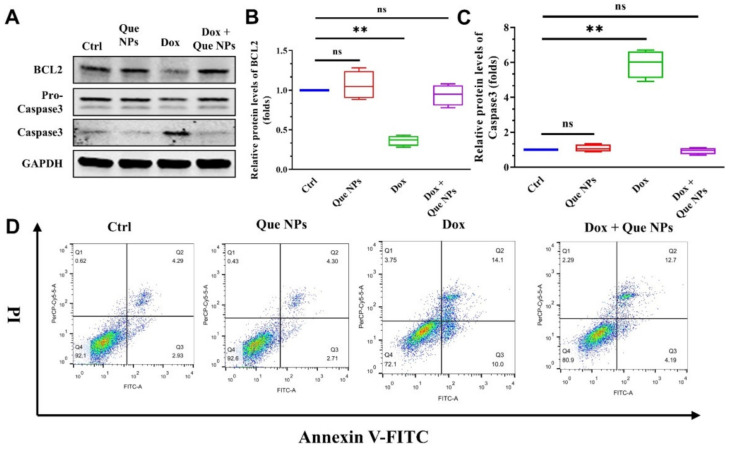
Evaluation of cellular apoptosis: (**A**) representative pictures of western blotting results; (**B**,**C**) statistical analysis of the expression changes of BCL2 (**B**) and Caspase3 (**C**) described in (**A**); and (**D**) flow cytometry analysis of the cellular apoptosis. **, *p* < 0.01; ns, no significance.

**Figure 4 pharmaceuticals-15-01524-f004:**
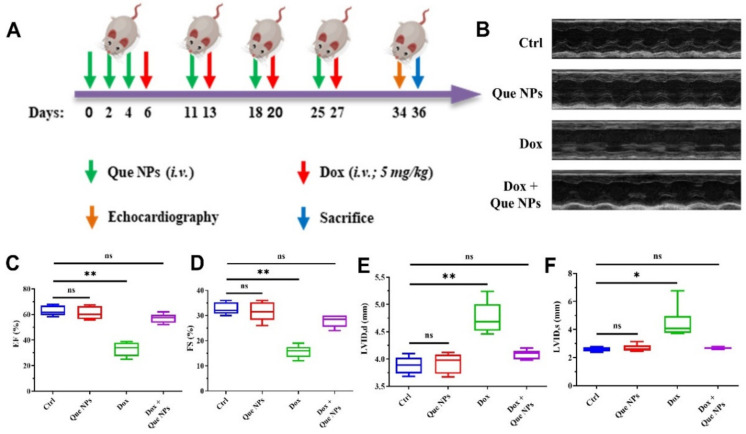
(**A**) Schematic description of the mouse model; (**B**) representative echocardiographic images; and (**C**–**F**) statistical analysis of the echocardiographic data. *, *p* < 0.05; **, *p* < 0.01; ns, no significance.

**Figure 5 pharmaceuticals-15-01524-f005:**
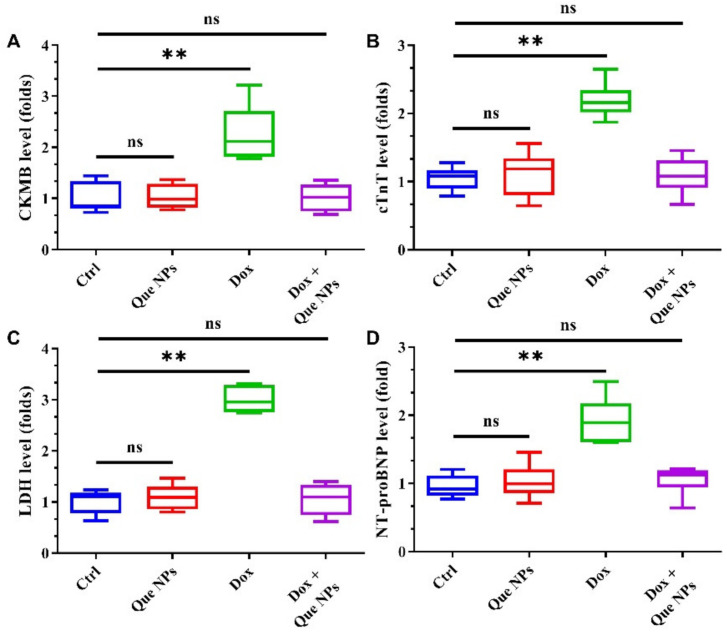
Cardiac injury assessment of Que NPs against cardiotoxicity induced by doxorubicin: (**A**–**D**) relative levels of cardiac injury protein markers CKMB (**A**), cTnT (**B**), LDH (**C**), NT-proBNP (**D**). **, *p* < 0.01; ns, no significance.

**Figure 6 pharmaceuticals-15-01524-f006:**
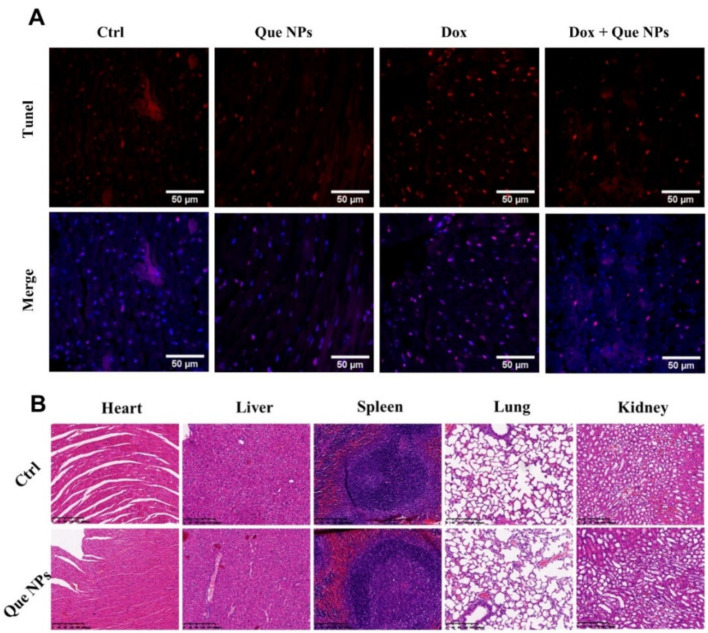
(**A**) Tunel staining of murine heart tissues; (**B**) representative HE staining pictures of murine major organs.

## Data Availability

The data which support the findings of this study are available from the corresponding author upon reasonable request.

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
