# Peer review of "Albumin-Based Zn (II)-Quercetin Enzyme Mimic Scavenging ROS for Protection against Cardiotoxicity Induced by Doxorubicin"

_pharmaceuticals, 2022, doi:10.3390/ph15121524_

Round 1

Reviewer 1 Report

Comments are indicated in the attached file.

Reviewer 2 Report

I've done my review of the manuscript entitled " Albumin-based Zn (II)-Quercetin Enzyme Mimic Scavenging ROS for Protection against Cardiotoxicity Induced by Doxorubicin by Zehua Shao and colleagues. 

In this manuscript, the authors reported the preparation of novel enzyme mimic via coordinated albumin, Zinc Ion (Zn2+), and quercetin, the enzyme mimics capable of simultaneously increasing the biocompatibility and it efficiently overcome the drawbacks of free quercetin achieved long circulation in vivo. They evaluated the efficacy of Que NPs scavenging free radicals for against cardiotoxicity induced by doxorubicin through cellular and animal models.

I find the topic very interesting however it needs minor revision. I would like to point out some issues with suggestions before publication.

Title: It’s fine

Abstract: The concern is that the authors failed to show a serious attitude toward the abstract format. The abstract must be more precise and must refer to the real results.

Introduction:  1. The introduction doesn't provide a good, generalized background of the topic that quickly gives the reader an appreciation.

2. To make the introduction more substantial, the author should provide several updated references to substantiate the claim made. 

Results and Discussion: Results are not clearly presented and is flawed and frustrating seriously. I dint notices characterization data
